# Healthy behaviors at age 50 years and frailty at older ages in a 20-year follow-up of the UK Whitehall II cohort: A longitudinal study

**Andres Gil-Salcedo**[1], **Aline Dugravot**[1], **Aurore Fayosse**[1], **Julien Dumurgier**[1], **Kim Bouillon**[2], **Alexis Schnitzler**[1], **Mika Kivimäki**[3,4], **Archana Singh-Manoux**[1,3], **Séverine Sabia**[1,3] *

1 Université de Paris, Inserm U1153, Epidemiology of Ageing and Neurodegenerative Diseases, France,
2 Département d'Information Médicale, Centre Hospitalier de Saint-Brieuc, Saint-Brieuc, France,
3 Department of Epidemiology and Public Health, University College London, London, United Kingdom,
4 Helsinki Institute of Life Sciences, University of Helsinki, Helsinki, Finland

* severine.sabia@inserm.fr

## Abstract

### Background

Frailty is associated with increased risk of various health conditions, disability, and death. Health behaviors are thought to be a potential target for frailty prevention, but the evidence from previous studies is based on older populations with short follow-ups, making results susceptible to reverse causation bias. We examined the associations of healthy behaviors at age 50, singly and in combination, as well as 10-year change in the number of healthy behaviors over midlife with future risk of frailty.

### Methods and findings

In this prospective cohort study of 6,357 (29.2% women; 91.7% white) participants from the British Whitehall II cohort, healthy behaviors—nonsmoking, moderate alcohol consumption, ≥2.5 hours per week of moderate to vigorous physical activity, and consumption of fruits or vegetables at least twice a day—were measured at age 50, and change in behaviors was measured between 1985 (mean age = 44.4) and 1997 (mean age = 54.8). Fried's frailty phenotype was assessed in clinical examinations in 2002, 2007, 2012, and 2015. Participants were classified as frail if they had ≥3 of the following criteria: slow walking speed, low grip strength, weight loss, exhaustion, and low physical activity. An illness–death model accounting for both competing risk of death and interval censoring was used to examine the association between healthy behaviors and risk of frailty. Over an average follow-up of 20.4 years (standard deviation, 5.9), 445 participants developed frailty. Each healthy behavior at age 50 was associated with lower risk of incident frailty: hazard ratio (HR) after adjustment for other health behaviors and baseline characteristics 0.56 (95% confidence interval [CI] 0.44–0.71; $p < 0.001$) in nonsmokers, 0.73 (95% CI 0.61–0.88; $p < 0.001$) for moderate alcohol consumption, 0.66 (95% CI 0.54–0.81; $p < 0.001$) for ≥2.5 hours of physical activity per week, and 0.76 (95% CI 0.59–0.98; $p = 0.03$) for consumption of fruits or vegetables at least

The Whitehall II data are available for sharing within the scientific community. Researchers can apply for data access at https://www.ucl.ac.uk/epidemiology-health-care/research/epidemiology-and-public-health/research/whitehall-ii/data-sharing.

**Funding:** The Whitehall II study has been supported by grants from the National Institute on Aging (R01AG056477, RF1AG062553, https://www.nih.gov/); the Medical Research Council UK (R024227, S011676, K013351, https://mrc.ukri.org/); the British Heart Foundation (RG/16/11/32334, https://www.bhf.org.uk/). MK was supported by the NordForsk, the Nordic Programme on Health and Welfare (https://www.nordforsk.org/programs/nordic-programme-health-and-welfare); the Academy of Finland (311492, https://www.aka.fi/en/), and Helsinki Institute of Life Science (https://www.helsinki.fi/en/helsinki-institute-of-life-science). SS is supported by the Agence Nationale de la Recherche (ANR-19-CE36-0004-01, https://anr.fr/en/anrs-role-in-research/missions/). The funders had no role in study design, data collection and analysis, decision to publish, or preparation of the manuscript.

**Competing interests:** The authors have declared that no competing interests exist.

**Abbreviations:** CES-D, Center for Epidemiology Studies Depression; CI, confidence interval; HDL, high-density lipoprotein; HR, hazard ratio; SD, standard deviation; STROBE, Strengthening the Reporting of Observational Studies in Epidemiology.

twice a day. A greater number of healthy behaviors was associated with reduced risk of frailty, with the HR for each additional healthy behavior being 0.69 (95% CI 0.62–0.76; $p < 0.001$) and the HR for having all versus no healthy behaviors at age 50 being 0.28 (95% CI 0.15–0.52; $p < 0.001$). Among participants with no or 1 healthy behavior in 1985, those who increased the number of healthy behaviors by 1997 were at a lower risk of frailty (mean follow-up = 16 years) compared with those with no such increase: the HR was 0.64 (95% CI 0.44–0.94; $p = 0.02$) for change to 2 healthy behaviors and 0.57 (95% CI 0.38–0.87; $p < 0.001$) for change to 3–4 healthy behaviors in 1997. The primary limitation of this study is potential selection bias during the follow-up due to missing data on frailty components.

## Conclusions

Our findings suggest that healthy behaviors at age 50, as well as improvements in behaviors over midlife, are associated with a lower risk of frailty later in life. Their benefit accumulates so that risk of frailty decreases with greater number of healthy behaviors. These results suggest that healthy behaviors in midlife are a good target for frailty prevention.

## Author summary

### Why was this study done?

- Frailty is a clinical syndrome associated with increased risk of several adverse health outcomes, including fracture, disability, and mortality.

- Health behaviors at older ages have been found to be associated with risk of frailty, but short follow-up in these studies raises the concern that the findings may reflect changes in health behaviors consequent to health-related conditions occurring in the years preceding frailty onset rather than a causal association between health behaviors and incident frailty.

### What did the researchers do and find?

- Data on smoking, alcohol consumption, physical activity, and fruits and vegetables consumption at age 50 were assessed among 6,357 participants of the Whitehall II study who were followed for incident frailty over 20.4 years.

- Frailty assessed at clinical examinations in 2002, 2007, 2012, and 2015 was defined as having 3 or more of the following criteria: slow walking speed, low grip strength, weight loss, exhaustion, and low physical activity.

- Each healthy behavior at age 50—nonsmoking, moderate alcohol consumption, practice of physical activity at least 2.5 hours per week, and consumption of fruits and vegetables at least twice a day—was associated with lower risk of frailty onset at older ages. In addition, participants with a greater number of healthy behaviors at age 50 had lower risk of frailty later in life, with those presenting all 4 healthy behaviors being at around 70% lower risk of developing frailty than those with none of these behaviors.

- Change in healthy behaviors between mean ages 44.4 and 54.8 years suggests that among individuals with no or only 1 healthy behavior, engagement in a greater number of healthy behaviors was associated with a reduced subsequent risk of frailty.

## What do these findings mean?

- Our findings suggest that health behaviors at age 50 are important determinants of frailty at older ages. We also found a dose–response association, such that the benefits for frailty are higher among those with a greater number of healthy behaviors.

- In our cohort, improvement in health behaviors over midlife was associated with reduced risk of developing frailty, suggesting that lifestyle interventions in midlife may be beneficial in frailty prevention.

## Introduction

The expected doubling of people aged 60 or older by 2050 [1] makes costs of health and social care a major challenge for the 21st century [2]. Prevention is seen to be important to tackle individual and societal consequences of age-related health conditions. Although the current healthcare system is mainly organized around organ-specific or single disease diagnosis, frailty is a holistic measure, reflecting an age-related syndrome characterized by vulnerability to stressors and declines in functioning of various physiological systems [3,4]. One in four persons aged 85 years or older is frail [4], and this condition is associated with an increased risk of multiple health outcomes, including cognitive disorders [5], fracture [6], disability [7], admission to hospital and long-term care [8], and death [9], making it an important target for prevention [4]. It is therefore important to identify modifiable risk factors to prevent or delay the onset of frailty [10–12].

Unhealthy behaviors, comprising smoking, excessive alcohol consumption, physical inactivity, and poor diet, are recognized risk factors for several chronic diseases [13] and mortality [14], but their importance for frailty remains unclear. Studies based on older population have reported that smokers [15,16], people less physically active [17–19], and those with poor diet [20,21] are at increased risk of frailty. For alcohol consumption, studies suggest a J-shaped association so that the lowest rates of frailty are observed among moderate drinkers [22–24]. Besides a few exceptions [24,25], previous studies examined health behaviors separately, often without controlling for other health behaviors. As health behaviors tend to cluster [26], the association observed for one behavior might be due to the presence of other unmeasured behaviors. In addition, most previous studies were limited to individuals aged 60 or older and were either cross-sectional or had a short follow-up, so it is not possible to rule out reverse causation bias as an explanation for observed associations [27]. Indeed, health behaviors may have been modified by health conditions that preceded frailty onset, such as chronic diseases or pre-frailty, an intermediate state before frailty [28].

To address these limitations, this analysis of the Whitehall II cohort study aimed to examine the association between health behaviors at age 50 and subsequent risk of frailty over a mean follow-up of 20 years. Each healthy behavior and the number of healthy behaviors were investigated to assess independent and cumulative associations with risk of frailty. A further (post

hoc) objective was to examine the association of change in the number of healthy behaviors over midlife and subsequent risk of frailty.

## Methods

This study is reported following the Strengthening the Reporting of Observational Studies in Epidemiology (STROBE) guideline (S1 STROBE Checklist). The study objectives and analysis plan were developed prior to data manipulation for an MSc internship (S1 Text). The analyses referred to as post hoc analyses were in response to suggestions from reviewers.

### Study population

Participants were drawn from the Whitehall II study, an ongoing cohort study established in 1985–1988 among 10,308 British civil servants aged 35–55 years at recruitment [29]. Data on sociodemographic, behavioral, anthropometric, and health-related conditions were collected at baseline and subsequent clinical follow-up examinations in 1991, 1997, 2002, 2007, 2012, and 2015, with each data collection wave taking approximately 2 years to be completed. Participant written, informed consent and research ethics approvals are renewed at each contact; the latest approval was by the NHS London—Harrow Research Ethics Committee, reference number 85/0938.

### Measures

**Health behaviors at age 50.**   Data from the first 4 data collection waves (in 1985, 1991, 1997, and 2002) were used to extract information on health behaviors at age 50 for each participant, allowing a 5-year margin. Health behaviors were assessed by questionnaire and categorized into 3 groups. Smoking status was categorized as "Never smoked," "Ex-smoking," and "Current smoking." The number of units of alcohol consumed in the last week was categorized as "None," "Moderate alcohol consumption" (1–14 units/week), and "High alcohol consumption" (>14 units per week) [30]. Physical activity was assessed as the number of hours of moderate to vigorous physical activity per week and categorized as "Inactive" (no moderate to vigorous physical activity per week), "Moderately active" (less than 2.5 hours per week), and "Active" (≥2.5 hours per week) [31]. Diet was assessed based on frequency of fruits and vegetables consumption and categorized as "Less than once a day," "Once a day," and "At least twice a day."

Exploratory analysis was undertaken to examine the shape of the association of continuous health behavior variables, alcohol consumption, and physical activity with onset of frailty (S1 and S2 Figs, respectively). Based on this exploratory analysis and findings in previous studies [32, 33], healthy behaviors were defined as follows: "Non-current smoking," "Moderate alcohol consumption" [23], "Physically active" [31], and "Consumption of fruits and vegetables at least twice a day." In addition, the number of healthy behaviors at age 50 was calculated for each participant and ranged between 0 (no healthy behaviors) and 4 (all healthy behaviors).

**Frailty.**   Frailty was measured at the clinical examination waves in 2002, 2007, 2012, and 2015 using the Fried's frailty phenotype [3,34]. Walking speed, grip strength, and weight loss were evaluated by trained nurses, and physical activity and exhaustion were assessed by questionnaire. The thresholds for these criteria were based on the original score [3]:

- Slow walking speed was defined as when the time spent walking 8 feet was ≥3.73 seconds for men (women) with height ≤173 (≤159) cm and ≥3.20 seconds for men (women) with height >173 (>159) cm.

- Low grip strength, assessed using a Smedley hand grip dynamometer, was defined for men as ≤29 kg for body mass index (BMI) ≤24 kg/m$^2$, ≤30 kg for BMI 24.1–28 kg/m$^2$, and ≤32 kg for BMI >28 kg/m$^2$. For women, low grip strength was defined as ≤17 kg for BMI ≤23 kg/m$^2$, ≤17.3 kg for BMI 23.1–26 kg/m$^2$, ≤18 kg for BMI 26.1–29 kg/m$^2$, and ≤21 kg for BMI >29 kg/m$^2$.

- Weight loss was defined as unintentional weight loss of 5% or more over the previous year according to Fried's criterion. Because weight was measured every 5 years, we used a cutoff of 10% of loss on body weight to define weight loss as used in the Women's Health Aging Study-I [35].

- Low physical activity was denoted by an energy expenditure of <383 kcal/week for men and <270 kcal/week for women, assessed based on responses to a questionnaire on frequency and duration of participation in 20 physical activities (e.g., cycling, housework, gardening activities). A metabolic equivalent value was assigned to each activity to calculate the energy expenditure of each participant.

- Exhaustion was defined based on responses to 2 items extracted from the Center for Epidemiology Studies Depression (CES-D) scale: "I felt that everything I did was an effort in the last week" and "I could not get going in the last week." If participants answered "occasionally or moderate amount of the time (3–4 days)" or "most or all of the time (5–7 days)" to either of these items, they were categorized as exhausted.

A frailty score was calculated as the number of the above criteria met, resulting in a score ranging from 0 to 5. Participants were considered as "frail" if their score was at 3 or higher [3].

**Mortality.**   Mortality data until August 2017 were drawn from the British national mortality register (National Health Services Central Registry). The tracing exercise was carried out using the National Health Service identification number of each participant.

**Covariates.**   Apart from sex, ethnicity, and education, which were assessed at baseline, covariates were drawn from the same wave as the measure of health behaviors at age 50 for each participant. Demographic factors included exact age, sex, ethnicity, and marital status ("married or cohabiting" versus "single, divorced or widowed"). Socioeconomic factors consisted of education (nonacademic qualification, high school, higher secondary, university, higher university degree) and occupational position (high, intermediate, and low; representing income and status at work). The number of morbidities at age 50 was calculated based on history of diabetes, coronary heart disease, stroke, chronic obstructive pulmonary disease, depression, arthritis, cancer, hypertension, and obesity using data from clinical examinations and electronic health records (the cancer registry, the National Hospital Episode statistics database, and the Mental Health Services Data Set, which in addition to in- and out-patient data, also has data on care in the community).

## Statistical analysis

Characteristics of participants were described by the number of healthy behaviors at age 50 and the frailty status at the end of the follow-up. Pearson's chi-squared test, Fisher's exact test, or chi-squared trend test was used to assess differences across categorical variables, and $t$ test and analyses of variance were used for continuous variables. In descriptive analysis, unadjusted frailty incidence rates per 1,000 person-years were calculated for each category of health behaviors. Participants contributed to person-years from the wave at which they were aged 50 to the first wave they were diagnosed as frail or their last clinical examination, whichever was first.

Frailty was assessed at clinical evaluations in 2002, 2007, 2012, and 2015, but the exact date of frailty onset was unknown (interval-censored data). In addition, some participants may have developed frailty and died between 2 data collection waves without being identified as frail, making death a competing event. We therefore used an interval-censored illness–death model with a Weibull distribution to assess hazard ratio (HR) of frailty with respect to health behavior categories. This analysis accounts for both the interval-censored nature of the data and competing risk of death [36].

The follow-up for all participants started at age 50 (±5), when health behaviors were assessed. The analysis used age as the timescale, and separate models for each health behavior, in 3 and then 2 categories, were serially adjusted for demographic factors and study wave in which follow-up began (model 1), socioeconomic factors (model 2), and the number of morbidities (model 3). The final model included all 4 health behaviors to assess their independent association with frailty onset (model 4). In further analyses, the number of healthy behaviors was entered into the models (models 1–3) as a categorical variable and then as a continuous variable to assess the HR of frailty associated with 1-point increment in the number of healthy behaviors.

Several sensitivity analyses were undertaken to assess robustness of the findings. First, we used an alternative definition of frailty based on meeting at least 2 of the Fried's criteria instead of 3. Second, as physical activity is included in both one of the exposures and the outcome, albeit coded differently, we redefined frailty without the "low physical activity" criterion and examined the impact of this possible bias on our results. Third, we used Cox regression instead of an illness–death model for comparison of results. Fourth, to take missing data into account, we used inverse probability weighting with information on the target population to calculate the probability of being included in the analytical sample using a logistic model that included demographic, socioeconomic, and behavioral factors at recruitment, morbidities and mortality over follow-up, and stepwise-selected interactions between covariates and health conditions. The inverse of these probabilities was used to weight results in the Cox regression.

We conducted several post hoc analyses in response to comments from reviewers. (1) Data on health behaviors assessed in 1985 and 1997 at mean ages 44.4 and 54.8 years were used to examine the association of change in health behaviors and subsequent frailty. The number of healthy behaviors was categorized into 3 groups (0–1, 2, and 3–4) at each time point, and change in these categories was examined in the analysis. Participants were followed for incident frailty from 1997 to the end of follow-up. Covariates in the analysis were assessed in 1997. For participants with missing data in 1997, data on health behaviors and covariates from 1991 or 2002 were used. (2) We examined the association between alternative definitions of healthy alcohol consumption and onset of frailty. (3) We undertook exploratory analysis on sedentary time as an additional health behavior. Sedentary time was assessed only in 1997 and could not be extracted at age 50; thus, we examined the association between sedentary time in 1997 and incident frailty. Sedentary time was measured as time spent sitting, calculated based on self-reported number of hours per week spent "sitting at work, driving, commuting or other" and "sitting at home e.g. watching TV, sewing, at desk," and was categorized into tertiles given the lack of definition of unhealthy sedentary time. A score of healthy behavior was calculated with all 5 behaviors, with healthy sedentary time defined as not being in the highest tertile of time spent sitting.

## Results

Of the 10,308 persons recruited to the Whitehall II study, 24 died before the age of 50 years, 337 were over 50 years at recruitment, and 6 were frail at age 50 (Fig 1), leading to a target

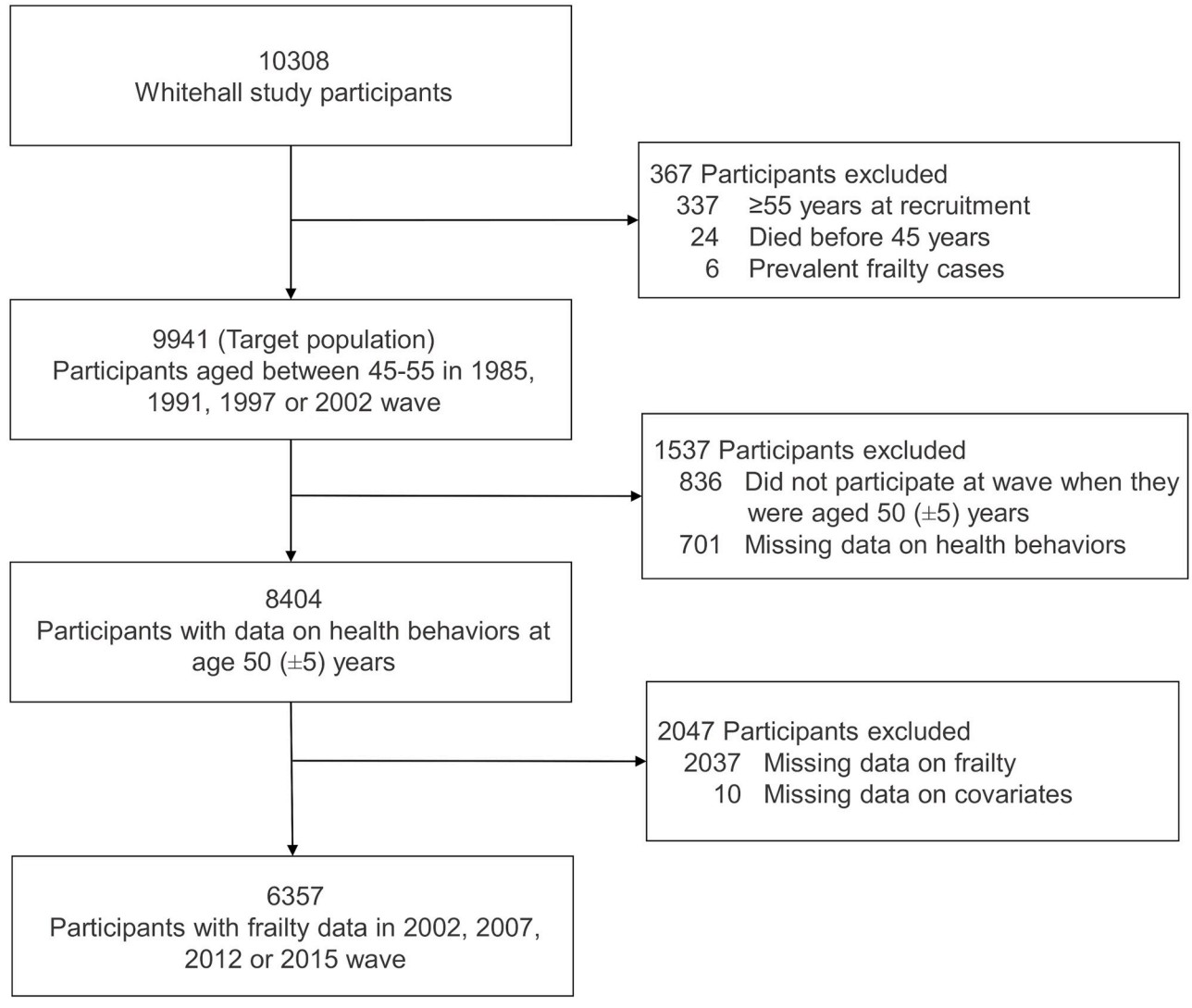

**Fig 1. Flowchart of sample selection.**

population of 9,941 participants. We excluded 836 participants who did not participate in any of the waves at which they were aged between 45 and 55 years, 701 participants with missing data on health behaviors, 10 with missing data on covariates, and 2,037 participants who had no frailty measures over the follow-up period. Thus, the final analytic cohort included 6,357 participants, of whom 445 (7%) became frail at a mean age of 72.1 (standard deviation [SD], 6.9) years and 701 (11%) died over a mean follow-up of 20.4 (SD, 5.9; range, 3.8–31) years. Further description of the composition of the analytic sample, corresponding to when participants were aged 50 (mean age = 50.4 [SD = 2.1] years), and corresponding mean follow-up are provided in S1 Table.

Table 1 shows the participants' characteristics according to frailty status at the end of follow-up. Women, nonwhite participants, those living alone, participants from a lower socioeconomic background, and those with a greater number of morbidities at age 50 were more likely to become frail over the follow-up (all $p < 0.001$). Details on prevalence of each morbidity at age 50 by frailty status at the end of follow-up are shown in S2 Table. In addition, frail

**Table 1. Characteristics of the study sample according to frailty status at the end of follow-up\*.**

| Characteristics at age 50 | Total study sample | Nonfrail | Frail | p |
|---|---|---|---|---|
| | (N = 6,357) | (N = 5,912) | (N = 445) | |
| Sex | | | | <0.001 |
| Men | 4,501 (70.8) | 4,261 (72.1) | 240 (53.9) | |
| Women | 1,856 (29.2) | 1,651 (27.9) | 205 (46.1) | |
| Ethnicity | | | | <0.001 |
| White | 5,830 (91.7) | 5,464 (92.4) | 366 (82.3) | |
| Nonwhite | 527 (8.3) | 448 (7.6) | 79 (17.7) | |
| Marital status | | | | <0.001 |
| Married/cohabiting | 4,942 (77.7) | 4,658 (78.8) | 284 (63.8) | |
| Single, divorced, or widowed | 1,415 (22.3) | 1,254 (21.2) | 161 (36.2) | |
| Education | | | | <0.001 |
| No academic qualification | 663 (10.4) | 590 (9.9) | 73 (16.4) | |
| High school | 2,062 (32.4) | 1,903 (32.2) | 159 (35.7) | |
| Higher secondary | 1,737 (27.3) | 1,638 (27.7) | 99 (22.2) | |
| University | 1,411 (22.2) | 1,319 (22.3) | 92 (20.7) | |
| Higher university degree | 484 (7.6) | 462 (7.8) | 22 (4.9) | |
| Occupational position | | | | <0.001 |
| Low | 845 (13.3) | 717 (12.1) | 128 (28.8) | |
| Intermediate | 2,824 (44.4) | 2,624 (44.4) | 200 (44.9) | |
| High | 2,688 (42.3) | 2,571 (43.5) | 117 (26.3) | |
| Number of morbidities | | | | <0.001 |
| 0 | 4,370 (68.7) | 4,113 (69.6) | 257 (57.7) | |
| 1 | 1,549 (24.4) | 1,408 (23.8) | 141 (31.7) | |
| 2 or more | 438 (6.9) | 391 (6.6) | 47 (10.6) | |
| Smoking status | | | | <0.001 |
| Never smoked | 3,202 (50.4) | 2,967 (50.2) | 235 (52.8) | |
| Ex-smoking | 2,378 (37.4) | 2,251 (38.1) | 127 (28.5) | |
| Current smoking | 777 (12.2) | 694 (11.7) | 83 (18.6) | |
| Alcohol consumption | | | | <0.001 |
| None | 989 (15.6) | 873 (14.8) | 116 (26.1) | |
| Moderate | 3,543 (56.2) | 3,318 (56.1) | 225 (50.6) | |
| High | 1,825 (28.7) | 1,721 (29.1) | 104 (23.4) | |
| Physical activity | | | | <0.001 |
| Inactive | 703 (11.1) | 599 (10.1) | 104 (23.4) | |
| Moderately active | 1,500 (23.6) | 1,358 (23.0) | 142 (31.9) | |
| Active | 4,154 (65.3) | 3,955 (66.9) | 199 (44.7) | |
| Fruits and vegetables consumption | | | | <0.001 |
| Less than once a day | 2,186 (34.4) | 2,007 (33.9) | 179 (40.2) | |
| Once a day | 2,445 (38.5) | 2,262 (38.3) | 183 (41.1) | |
| At least twice a day | 1,726 (27.2) | 1,643 (27.8) | 83 (18.6) | |
| **Frailty components at the end of follow-up[†]** | | | | |
| Slow walking speed | 503 (7.9) | 230 (3.9) | 273 (61.3) | <0.001 |
| Low grip strength | 1,399 (22.0) | 1,075 (18.2) | 324 (72.8) | <0.001 |
| Exhaustion | 794 (12.5) | 511 (8.6) | 283 (63.6) | <0.001 |
| Low physical activity | 2,210 (34.8) | 1,799 (30.4) | 411 (92.4) | <0.001 |

*(Continued)*

**Table 1.** (Continued)

| Characteristics at age 50 | Total study sample | Nonfrail | Frail | p |
|---|---|---|---|---|
| | (*N* = 6,357) | (*N* = 5,912) | (*N* = 445) | |
| Weight loss | 302 (4.7) | 181 (3.1) | 121 (27.2) | <0.001 |

*Values are numbers (percentages). Percentages are reported in column.

†End of follow-up corresponds to date of frailty diagnosis or last wave of clinical examination for nonfrail participants.

participants at older age were more likely to be current smokers, alcohol abstainers, physically inactive, and to eat fruits and vegetables less frequently at age 50 (all $p < 0.001$). At the end of follow-up, each frailty component, apart from weight loss, was present in over 60% of the frail group (Table 1). The proportion of participants who were white, married/cohabiting, with higher socioeconomic background, and with fewer morbidities was higher among those with a greater number of healthy behaviors at age 50 (all $p < 0.05$; S3 Table).

Because there was no evidence of an interaction of the number of healthy behaviors with sex ($p = 0.22$), education ($p = 0.47$), and occupational position ($p = 0.14$), analyses were undertaken in the total population. The association between each health behavior at age 50, categorized in 3 groups, and subsequent risk of frailty is shown in Table 2. Unadjusted incidence rates per 1,000 person-years among never, ex-, and current smokers were 3.57, 2.60, and 5.41, respectively. In analyses adjusted for demographic factors, never (HR 0.59, 95% confidence

**Table 2. Association between health behaviors at age 50 and onset of frailty over a mean follow-up of 20 years.**

| Health behaviors | N frail/N total | Frail % | Model 1*† | | Model 2*‡ | | Model 3*§ | | Model 4‖ | |
|---|---|---|---|---|---|---|---|---|---|---|
| | | | HR (95% CI) | p | HR (95% CI) | p | HR (95% CI) | p | HR (95% CI) | p |
| Smoking status | | | | | | | | | | |
| Never smoked | 235/3,202 | 7.34 | 0.59 (0.46–0.77) | <0.001 | 0.62 (0.48–0.81) | <0.001 | 0.62 (0.47–0.80) | <0.001 | 0.68 (0.52–0.89) | 0.01 |
| Ex-smoking | 127/2,378 | 5.34 | 0.50 (0.38–0.67) | <0.001 | 0.52 (0.39–0.69) | <0.001 | 0.50 (0.37–0.66) | <0.001 | 0.53 (0.40–0.71) | <0.001 |
| Current smoking | 83/777 | 10.68 | 1 (ref) | | 1 (ref) | | 1 (ref) | | 1 (ref) | |
| Alcohol consumption | | | | | | | | | | |
| None | 116/989 | 11.73 | 1.22 (0.91–1.65) | 0.18 | 1.08 (0.79–1.46) | 0.64 | 1.11 (0.81–1.51) | 0.51 | 1.15 (0.85–1.57) | 0.37 |
| Moderate | 225/3,543 | 6.35 | 0.76 (0.59–0.97) | 0.03 | 0.72 (0.56–0.92) | 0.01 | 0.75 (0.58–0.97) | 0.03 | 0.76 (0.59–0.98) | 0.03 |
| High | 104/1,825 | 5.70 | 1 (ref) | | 1 (ref) | | 1 (ref) | | 1 (ref) | |
| Physical activity | | | | | | | | | | |
| Inactive | 104/703 | 14.79 | 1 (ref) | | 1 (ref) | | 1 (ref) | | 1 (ref) | |
| Moderately active | 142/1,500 | 9.47 | 0.81 (0.62–1.05) | 0.11 | 0.87 (0.65–1.17) | 0.37 | 0.90 (0.68–1.19) | 0.47 | 0.94 (0.71–1.25) | 0.67 |
| Active | 199/4,154 | 4.79 | 0.55 (0.42–0.72) | <0.001 | 0.59 (0.45–0.78) | <0.001 | 0.61 (0.47–0.81) | <0.001 | 0.66 (0.48–0.88) | 0.001 |
| Fruits and vegetables consumption | | | | | | | | | | |
| Less than once a day | 179/2,186 | 8.19 | 1 (ref) | | 1 (ref) | | 1 (ref) | | 1 (ref) | |
| Once a day | 183/2,445 | 7.48 | 0.83 (0.67–1.02) | 0.08 | 0.87 (0.70–1.08) | 0.19 | 0.85 (0.68–1.06) | 0.14 | 0.91 (0.73–1.13) | 0.40 |
| At least twice a day | 83/1,726 | 4.81 | 0.64 (0.49–0.85) | <0.001 | 0.68 (0.52–0.90) | 0.01 | 0.65 (0.50–0.87) | <0.001 | 0.70 (0.53–0.92) | 0.01 |

*Models 1, 2, and 3 were estimated for each health behavior separately.

†Model 1: age as a timescale, adjusted for sex, ethnicity, marital status, and wave of inclusion.

‡Model 2: model 1 additionally adjusted for education and occupational position.

§Model 3: model 2 additionally adjusted for the number of morbidities at age 50.

‖Model 4: model 3 additionally adjusted for all other health behaviors.

Abbreviations: CI, confidence interval; HR: hazard ratio; ref, reference group

**Table 3. Association between healthy behaviors at age 50 in 2 categories and onset of frailty over a mean follow-up of 20 years.**

| Healthy behaviors | N frail/N total | Frail % | Model 1[*][†] | | Model 2[*][‡] | | Model 3[*][§] | | Model 4[‖] | |
|---|---|---|---|---|---|---|---|---|---|---|
| | | | HR (95% CI) | p | HR (95% CI) | p | HR (95% CI) | p | HR (95% CI) | p |
| Noncurrent smoking | | | | | | | | | | |
| No | 83/777 | 10.68 | 1 (ref) | | 1 (ref) | | 1 (ref) | | 1 (ref) | |
| Yes | 362/5,580 | 6.49 | 0.56 (0.44–0.72) | <0.001 | 0.58 (0.45–0.75) | <0.001 | 0.57 (0.44–0.73) | <0.001 | 0.56 (0.44–0.71) | <0.001 |
| Moderate alcohol consumption | | | | | | | | | | |
| No | 220/2,814 | 7.82 | 1 (ref) | | 1 (ref) | | 1 (ref) | | 1 (ref) | |
| Yes | 225/3,543 | 6.35 | 0.68 (0.56–0.82) | <0.001 | 0.69 (0.57–0.84) | <0.001 | 0.71 (0.59–0.86) | 0.001 | 0.73 (0.61–0.88) | <0.001 |
| Physical activity | | | | | | | | | | |
| No | 246/2,003 | 12.28 | 1 (ref) | | 1 (ref) | | 1 (ref) | | 1 (ref) | |
| Yes | 199/4,154 | 4.79 | 0.63 (0.51–0.78) | <0.001 | 0.64 (0.52–0.79) | <0.001 | 0.66 (0.53–0.81) | <0.001 | 0.66 (0.54–0.81) | <0.001 |
| Fruits and vegetables consumption at least twice a day | | | | | | | | | | |
| No | 362/4,631 | 7.82 | 1 (ref) | | 1 (ref) | | 1 (ref) | | 1 (ref) | |
| Yes | 83/1,726 | 4.81 | 0.71 (0.56–0.92) | 0.01 | 0.74 (0.57–0.95) | 0.02 | 0.72 (0.55–0.94) | 0.02 | 0.76 (0.59–0.98) | 0.03 |

[*]Models 1, 2, and 3 were estimated for each healthy behavior separately.

[†]Model 1: age as a timescale, adjusted for sex, ethnicity, marital status, and wave of inclusion.

[‡]Model 2: model 1 additionally adjusted for education and occupational position.

[§]Model 3: model 2 additionally adjusted for the number of morbidities at age 50.

[‖]Model 4: model 3 additionally adjusted for all other healthy behaviors.

Abbreviations: CI, confidence interval; HR, hazard ratio; ref, reference group

interval [CI] 0.46–0.77; $p < 0.001$) and ex-smokers (HR 0.50, 95% CI 0.38–0.67; $p < 0.001$) had a lower risk of incident frailty compared with current smokers. Higher unadjusted incidence rate of frailty per 1,000 person-years was found among alcohol abstainers compared with moderate and high alcohol drinkers (5.80, 3.04, and 2.90, respectively; $p < 0.001$); the incidence rate did not differ between moderate and high alcohol drinkers ($p = 0.68$). In the analysis adjusted for demographic factors, moderate alcohol consumption was associated with a reduced risk of frailty compared with high alcohol consumption (HR 0.76, 95% CI 0.59–0.97; $p = 0.03$), but alcohol abstainers did not have a different risk of frailty compared with high alcohol consumption (HR 1.22, 95% CI 0.91–1.65; $p = 0.18$). Regarding physical activity, only active participants had a lower risk of frailty compared with inactive participants (unadjusted incidence rate per 1,000 person-years, 2.44 versus 6.78, respectively; HR adjusted for demographic factors 0.55, 95% CI 0.42–0.72; $p < 0.001$). A reduced risk of frailty was observed in participants who consumed fruits and vegetables at least twice a day compared with those who consumed fruits and vegetables less than once per day (unadjusted incidence rate per 1,000 person-years, 2.25 versus 3.95, respectively; HR adjusted for demographic factors 0.64, 95% CI 0.49–0.85; $p < 0.001$). These associations were slightly attenuated after further adjustment for socioeconomic factors, morbidities, and mutual adjustment for health behaviors (Table 2).

Subsequent analyses of each healthy behavior as binary variables (Table 3) showed lower risk of frailty for noncurrent smoking (HR 0.56; 95% CI 0.44–0.72; $p < 0.001$), moderate alcohol consumption (HR 0.68; 95% CI 0.56–0.82; $p < 0.001$), being physically active (HR 0.63; 95% CI 0.51–0.78; $p < 0.001$), and fruits and vegetables consumption at least twice a day (HR 0.71; 95% CI 0.56–0.92; $p = 0.01$) in the analysis adjusted for demographic variables. Associations were slightly reduced after further adjustments for covariates.

Lower risk of incident frailty was observed in participants who had 2 or more healthy behaviors at age 50 compared with those who had none (Fig 2). In the analysis adjusted for

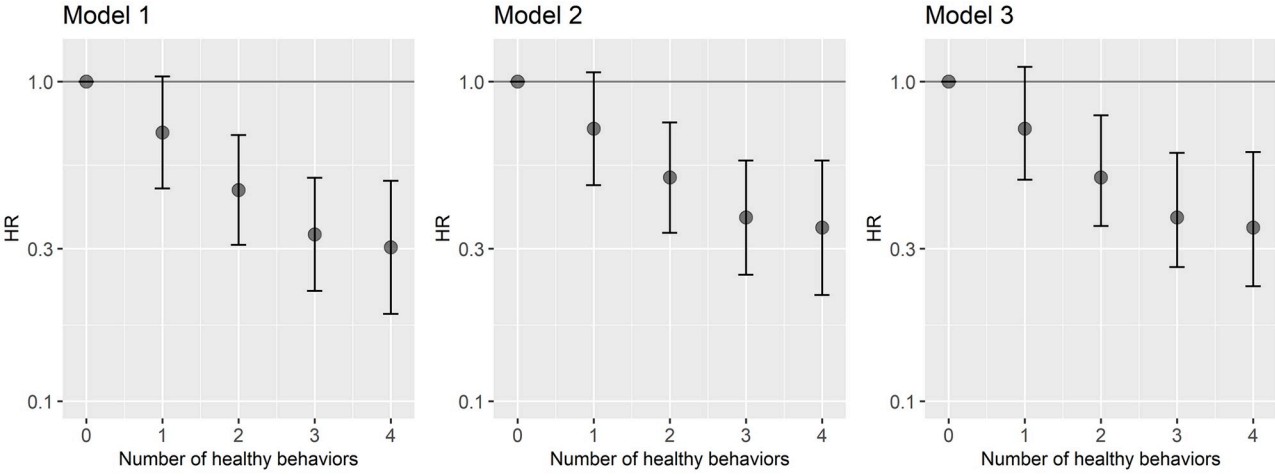

**Fig 2. Association between the number of healthy behaviors at age 50 and the onset of frailty over a mean follow-up of 20 years.** Model 1: age as a timescale, adjusted for sex, ethnicity, marital status, and wave at inclusion. Model 2: model 1 additionally adjusted for education and occupational position. Model 3: model 2 additionally adjusted for the number of morbidities at age 50. Associated estimations are in S4 Table. HR, hazard ratio.

demographic and socioeconomic variables and the number of morbidities at age 50 (S4 Table), compared with no healthy behaviors at age 50, the HRs of incident frailty were 0.74 (95% CI 0.45–1.21; $p = 0.22$) for 1 healthy behavior, 0.47 (95% CI 0.29–0.77; $p = 0.01$) for 2, 0.32 (95% CI 0.19–0.53; $p < 0.001$) for 3, and 0.28 (95% CI 0.15–0.52; $p < 0.001$) for all 4 healthy behaviors. A linear association was observed between the number of healthy behaviors at age 50 and risk of subsequent frailty (Wald test $p < 0.001$), the HR per 1 additional healthy behavior being 0.69 (95% CI 0.62–0.76; $p < 0.001$).

## Sensitivity analysis

All sensitivity analyses yielded results that were similar to those in the main analyses so that the risk of frailty decreased as the number of healthy behaviors at age 50 increased (S3 Fig). In the analysis based on 1,673 frailty cases defined as having at least 2 instead of 3 of Fried's frailty criteria, the HR of frailty per 1 additional healthy behavior was 0.74 (95% CI 0.70–0.78; $p < 0.001$). When frailty was defined as having 2 of the Fried's criteria, excluding the "low physical activity" criterion, 694 cases were identified, and the HR of frailty was 0.75 (95% CI 0.70–0.82; $p < 0.001$) for each additional healthy behavior. Compared with the illness–death model, Cox regression showed similar findings (HR 0.68; 95% CI 0.62–0.76; $p < 0.001$; S5 Table). Finally, compared with the 6,357 participants retained in the study sample, participants excluded from the analysis ($N = 3,584$) were more likely to be older (excluded versus included participants' mean age: 45.5 versus 44.2 years), women (39.3 versus 29.2%), and nonwhite (13.7 versus 8.3%) and to have a lower occupational position at age 50 (33.9 versus 13.3%, all $p < 0.001$), leading us to assess the role of missing data in the analysis. Results based on inverse probability weighting (S5 Table) showed a similar risk reduction of frailty for each supplementary healthy behavior (HR 0.70, 95% CI 0.63–0.78; $p < 0.001$).

## Post hoc analyses

Change in the number of healthy behaviors was calculated among 6,435 participants with data on health behaviors available in 1985 and 1997 ($N = 5,380$) (or 1991 [$N = 905$] or 2002 [$N = 150$] when data in 1997 were missing). This corresponds to change over a mean of 10.4

**Table 4. Association between change in the number of healthy behaviors over midlife and onset of frailty over a mean follow-up of 16 years.**

| Number of healthy behaviors | | N frail/N total | Frail % | Model 1[†] | | Model 2[‡] | | Model 3[§] | |
|---|---|---|---|---|---|---|---|---|---|
| In 1985, mean age = 44.4 years | In 1997*, mean age = 54.8 years | | | HR (95% CI) | p | HR (95% CI) | p | HR (95% CI) | p |
| 0–1 | 0–1 | 66/422 | 15.64 | 1 (ref) | | 1 (ref) | | 1 (ref) | |
| 0–1 | 2 | 51/555 | 9.19 | 0.61 (0.42–0.90) | 0.01 | 0.63 (0.43–0.92) | 0.02 | 0.64 (0.44–0.94) | 0.02 |
| 0–1 | 3–4 | 38/426 | 8.92 | 0.56 (0.37–0.85) | 0.007 | 0.57 (0.37–0.87) | 0.009 | 0.57 (0.38–0.87) | 0.009 |
| 2 | 0–1 | 25/248 | 10.08 | 0.65 (0.41–1.04) | 0.71 | 0.66 (0.41–1.05) | 0.08 | 0.69 (0.43–1.10) | 0.12 |
| 2 | 2 | 84/999 | 8.41 | 0.54 (0.38–0.75) | <0.001 | 0.54 (0.39–0.76) | <0.001 | 0.57 (0.41–0.81) | 0.001 |
| 2 | 3–4 | 79/1,329 | 5.94 | 0.35 (0.24–0.50) | <0.001 | 0.36 (0.25–0.51) | <0.001 | 0.37 (0.26–0.53) | <0.001 |
| 3–4 | 0–1 | 7/72 | 9.72 | 0.66 (0.29–1.48) | 0.31 | 0.67 (0.29–1.52) | 0.34 | 0.56 (0.25–1.27) | 0.17 |
| 3–4 | 2 | 29/545 | 5.32 | 0.39 (0.25–0.62) | <0.001 | 0.39 (0.25–0.63) | <0.001 | 0.41 (0.26–0.65) | <0.001 |
| 3–4 | 3–4 | 85/1,839 | 4.62 | 0.29 (0.21–0.42) | <0.001 | 0.30 (0.21–0.43) | <0.001 | 0.32 (0.23–0.46) | <0.001 |

*Or in 1991 (N = 905) or in 2002 (N = 150) if missing data in 1997.

[†]Model 1: age as timescale, adjusted for sex, ethnicity, marital status, and wave of second measurement of health behaviors.

[‡] Model 2: model 1 additionally adjusted for education and occupational position.

[§] Model 3: model 2 additionally adjusted for the number of morbidities at the second measurement of health behaviors.

Abbreviations: CI, confidence interval; HR, hazard ratio; ref, reference

(SD = 2.4) years, with mean age being 44.4 (SD = 5.9) years at the first measure and 54.8 (SD = 6.4) years at the second measure. The mean number of healthy behaviors increased by 0.4 (SD = 1.0) during this 10-year period. In a fully adjusted illness–death model, with no or 1 healthy behavior in both 1985 and 1997 as the reference group, HR for 2 healthy behaviors at both measures was 0.57 (0.41–0.81; p = 0.001) and 0.32 (0.23 to 0.46; p < 0.001; Table 4) for 3–4 healthy behaviors at both measures. Among those with no or 1 healthy behavior in 1985, change to 2 (HR 0.64 95% CI 0.44–0.94; p = 0.02) or 3–4 (0.57 95% CI 0.38–0.87; p = 0.009) healthy behaviors in 1997 was associated with lower risk of frailty compared with 0 or 1 healthy behavior in 1997 (Table 4).

Associations between alternative definitions of healthy alcohol consumption and frailty onset are presented in S6 Table. In fully adjusted models, as compared with "moderate or high alcohol consumption," "no alcohol consumption" was associated with higher risk of frailty (HR 1.36, 95% CI 1.08–1.70; p = 0.008), and there was no statistically significant difference in risk of frailty between "no or moderate alcohol consumption" and "high alcohol consumption" (HR 0.83, 95% CI 0.66–1.06; p = 0.14).

Post hoc analysis on the association between sedentary time in 1997 and incident frailty based on 5,296 participants (N incident frailty = 351) is presented in S7 Table. In the analysis adjusted for demographic factors, sitting time in the first (<2.8 hours per day; HR 0.74, 95% CI 0.57–0.96; p = 0.02) and the second (2.8–4.4 hours per day; HR 0.77, 95% CI 0.59–1.00; p = 0.05) tertiles was associated with lower risk of frailty compared with sitting time in the top tertile (>4.4 hours per day). In the fully adjusted analysis (including other healthy behaviors in 1997), the HR of frailty associated with healthy sitting time (<4.4 hours per day) was 0.79 (95% CI 0.63–0.99; p = 0.04). When healthy sitting time was included in the score of healthy

behaviors (range 0–5), HR per 1 additional healthy behavior was 0.72 (95% CI 0.65–0.81; $p < 0.001$).

## Discussion

In this longitudinal study based on over 6,000 participants, healthy behaviors—defined as not smoking, moderate alcohol consumption, at least 2.5 hours per week of moderate to vigorous physical activity, and consumption of fruits and vegetables at least twice a day—assessed at age 50 were independently associated with a reduced risk of frailty onset over a mean 20-year follow-up, with risk reduction ranging from 24% for fruits and vegetables consumption at least twice a day to 44% for nonsmoking. The benefits of healthy behaviors were cumulative, as the risk of frailty decreased progressively with greater number of healthy behaviors to reach a risk reduction of 72% among those with all 4 healthy behaviors at age 50 compared with those without any healthy behaviors at that age. Our results also show that favorable change in the number of healthy behaviors in midlife was associated with a reduced risk of subsequent frailty compared with persistently low number of healthy behaviors.

### Comparison with previous studies

Previous studies have suggested that current nonsmoking (including never and ex-smokers) [16], moderate to vigorous physical activity [17,18], moderate alcohol intake [22,23], and good dietary patterns [20,21] are associated with lower risk of frailty. To our knowledge, most previous studies examined one behavior at a time, some of which took into account smoking and, to a lesser extent, alcohol consumption but did not adjust for other health behaviors, despite known clustering of behaviors [26]. One notable exception is a study of an older population followed for 3.3 years that reported an association between alcohol consumption and lower risk of frailty, taking into account smoking, physical activity, and diet [24]. Our findings suggest an independent association of each of the 4 health behaviors examined with risk of subsequent frailty, in agreement with previous findings but addressing the possible biases of confounding and reverse causation. Our findings highlight the potential key role of healthy behaviors in midlife for frailty prevention later in life, with risk reduction ranging from 24% to 44% depending on the health behavior considered. These associations were adjusted for socio-economic factors, but it is worth noting that health behaviors also contribute to the association between socioeconomic advantage and frailty, as shown in previous analysis from the Whitehall II cohort study [37].

   In addition to the independent association of each health behavior with the risk of frailty, the present study suggests that their benefits have a cumulative effect so that for each additional healthy behavior at age 50, there was a 31% risk reduction in frailty. These results are in accordance with a recent study based on 1,309 participants from the general population (mean age at health behaviors measure = 70 years) showing the number of favorable health behaviors among nonsmoking, physically active, healthy diet, adequate sleeping duration, not being sedentary, and daily social interaction to be associated with lower incidence of self-reported frailty symptoms assessed twice over an 8-year period [25]. Our findings extend current knowledge on the cumulative role of behavioral factors also found for other aging outcomes, including chronic diseases [32,38], cognitive and physical function [33,39], quality of life [40], disability [41], and mortality [14]. The present study, using repeated measures of health behaviors, also adds to the current literature by suggesting that favorable changes in lifestyle during midlife might reduce the risk of frailty at older ages.

   Multiple mechanisms may explain the association between healthy behaviors and frailty. Physical activity, moderate alcohol intake, and healthy diet are associated with increase in

adiponectin and high-density lipoprotein (HDL) levels [42–44]. Alcohol intake, physical activity, fruits and vegetables consumption, and smoking may play a role in oxidative stress regulation [45–48]. Levels of adiponectin, HDL, and oxidative stress, in turn, correlate with inflammation processes and are suggested to be involved in frailty development [49,50]. Furthermore, unhealthy behaviors are associated with higher incidence of multiple chronic diseases [13] known to increase the risk of frailty [28], making chronic conditions potential intermediaries of the association between health behaviors and frailty onset. Indeed, in the present study, 10.6% of participants who developed frailty during the follow-up had at least 2 morbidities already at age 50, and this proportion reached 47.9% at the time of frailty onset.

## Strengths and limitations

The present study has several strengths, including the measurement of health behaviors for all participants at age 50 (SD, 2.1 years) and a mean follow-up duration of 20 years. In contrast, most previous studies were undertaken among older populations, with a short follow-up for the onset of frailty. As the frailty phenotype develops over several years and includes an intermediate state of prefrailty [3], the possibility that health behaviors are modified because of health conditions related to prefrailty status in these studies cannot be excluded. The design of the present study minimizes reverse causation bias of this kind because health behaviors were measured at age 50 and thus before the development of frailty. In addition, the use of an interval-censored illness–death model in our analyses allowed us to take into account the competing risk of death and the imprecision in the date of frailty onset due to intermittent ascertainment of frailty status over the data collection waves.

Our findings need to be considered in light of the study limitations. First, the population composed of British civil servants is not representative of the general population. However, this is an unlikely source of bias because previous analyses show that although participants from the cohort are healthier in terms of risk factor levels and prevalence of cardiovascular disease, the associations between cardiovascular risk factors, including health behaviors, and cardiovascular disease are similar to those in general population studies [51]. Another limitation concerns missing data arising from nonparticipation in clinical examinations during the 2 decades of follow-up. We undertook sensitivity analyses using inverse probability weighting to assess the role of missing data on our findings, and results from these analyses were consistent with the main findings. Third, the use of a simple measure such as fruits and vegetables consumption to assess dietary pattern may not adequately reflect the association between diet and frailty. Fourth, the use of observational data cannot rule out the role of unmeasured confounders. Finally, disability status at age 50 was not available, but because all participants were in full-time employment, the likelihood of confounding bias by disability might be small.

## Clinical implications and future research

In the context of aging of populations worldwide, effective prevention is key in order to allow older adults to remain healthy as long as possible and reduce the societal burden of aging. This includes prevention of frailty, a geriatric syndrome associated with higher risk of several health conditions and increased healthcare needs. Frailty is more prevalent among women and participants from the lower socioeconomic group; in our analyses, the number of healthy behaviors at age 50 and the change in this number over midlife were similarly associated with risk of incident frailty in both sexes and in different socioeconomic groups. Because health behaviors are modifiable, they are a good target for frailty prevention at the population level. The present findings along with those from previous studies support the development of interventions in clinical settings to encourage a healthy lifestyle as a whole rather than

focusing on specific healthy behaviors to promote healthy aging. Among the multiple tools to assess frailty, we chose to use Fried's frailty phenotype given its robust association with health outcomes; its use in several studies, making our results amenable to replication; and the fact that its association with health outcomes such as mortality has been shown to be similar to that of more extensive measures such as the frailty index based on the accumulation of deficits model [52]. Research on health behaviors would benefit from more precise and objective measures because much of the research, including ours, remains based on self-reported behaviors. For example, accelerometers are increasingly being used for objective assessment of physical activity and sedentary behavior [53]. A further area of research is better assessment of sedentary behavior; our post hoc analysis suggests an independent role of sedentary behavior on frailty onset.

## Conclusion

Our analysis based on an observational cohort study showed that a greater number of healthy behaviors at age 50 as well as improvement in health behaviors over midlife were associated with lower risk of frailty over a 20-year follow-up period. These findings highlight the importance of healthy behavior in midlife for prevention of frailty at older ages.

## Supporting information

**S1 Table. Wave of extraction of health behaviors at age 50.**
(DOCX)

**S2 Table. Chronic conditions at age 50 according to frailty status at the end of follow-up.**
(DOCX)

**S3 Table. Characteristics of the study sample according to the number of healthy behaviors at age 50.**
(DOCX)

**S4 Table. Association between the number of healthy behaviors at age 50 and the risk of frailty over a 20-year follow-up.** CI, confidence interval; HR, hazard ratio.
(DOCX)

**S5 Table. Association between the number of healthy behaviors at age 50 and risk of frailty using Cox regression only and with inverse probability weighting.** CI, confidence interval; HR, hazard ratio.
(DOCX)

**S6 Table. Association between different definitions of healthy alcohol consumption at age 50 and onset of frailty over a mean follow-up of 20 years.** CI, confidence interval; HR, hazard ratio.
(DOCX)

**S7 Table. Association between sedentary behaviors in 1997 (mean age = 55.5 [SD = 6.0] years) and onset of frailty over a mean follow-up of 16 years, $N$ = 5,296.** CI, confidence interval; HR, hazard ratio.
(DOCX)

**S1 Fig. Association between units of alcohol per week at age 50 and incident frailty using cubic spline regression.** Blue line represents the hazard ratio for frailty compared with "14 units of alcohol per week" (reference), and black dashed line represents the corresponding

95% confidence interval estimated based on a Cox regression model with age at timescale adjusted for sex, ethnicity, marital status, and wave at inclusion.
(TIF)

**S2 Fig. Association between number of hours of moderate and vigorous physical activity at age 50 and incident frailty using cubic spline regression.** Blue line represents the hazard ratio for frailty compared with 2.5 hours of moderate and vigorous physical activity (reference), and black dashed line represents the corresponding 95% confidence interval estimated based on a Cox regression model with age at timescale adjusted for sex, ethnicity, marital status, and wave at inclusion.
(TIF)

**S3 Fig. Sensitivity analysis.** Age was used as timescale, and models are adjusted for sex, ethnicity, marital status, wave of inclusion, education, occupational position, and number of morbidities at age 50. HR, hazard ratio.
(TIF)

**S1 Text. Prospective analysis plan.**
(DOCX)

**S1 STROBE Checklist. STROBE checklist for cohort studies.** STROBE, Strengthening the Reporting of Observational Studies in Epidemiology.
(DOCX)

# Acknowledgments

We are grateful to the participants from civil service departments and their welfare, personnel, and establishment officers and all members of the Whitehall II study teams.

The lead author affirms that this manuscript is an honest, accurate, and transparent account of the study being reported; that no important aspects of the study have been omitted; and that any discrepancies from the study as planned have been explained. The lead author in this statement is the study guarantor.

# Author Contributions

**Conceptualization:** Aline Dugravot, Archana Singh-Manoux, Séverine Sabia.

**Data curation:** Andres Gil-Salcedo, Aline Dugravot, Aurore Fayosse, Kim Bouillon.

**Formal analysis:** Andres Gil-Salcedo, Aline Dugravot.

**Funding acquisition:** Mika Kivimäki, Archana Singh-Manoux.

**Investigation:** Andres Gil-Salcedo, Aline Dugravot, Aurore Fayosse, Julien Dumurgier, Alexis Schnitzler, Mika Kivimäki, Archana Singh-Manoux, Séverine Sabia.

**Methodology:** Aline Dugravot, Séverine Sabia.

**Supervision:** Séverine Sabia.

**Validation:** Aline Dugravot, Séverine Sabia.

**Visualization:** Andres Gil-Salcedo, Aline Dugravot, Séverine Sabia.

**Writing – original draft:** Andres Gil-Salcedo, Séverine Sabia.

**Writing – review & editing:** Andres Gil-Salcedo, Aline Dugravot, Aurore Fayosse, Julien Dumurgier, Kim Bouillon, Alexis Schnitzler, Mika Kivimäki, Archana Singh-Manoux, Séverine Sabia.

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
