## [Decision Letter · Decision Letter 0]

6 Feb 2020

Dear Dr. Sabia,

Thank you very much for submitting your manuscript "Association between healthy behaviors at age 50 and frailty at older ages: 20-year follow-up of the Whitehall II cohort study" (PMEDICINE-D-19-04222) for consideration at PLOS Medicine. 

Your paper was discussed among the editorial team, evaluated by an academic editor with relevant expertise, and sent to independent reviewers, including a statistical reviewer. The reviews are appended at the bottom of this email and any accompanying reviewer attachments can be seen via the link below:

[LINK]

In light of these reviews, we will not be able to accept the manuscript for publication in the journal in its current form, but we would like to invite you to submit a revised version that fully addresses the reviewers' and editors' comments. You will appreciate that we cannot make a decision about publication until we have seen the revised manuscript and your response, and we expect to seek re-review by one or more of the reviewers. 

We hope to receive your revised manuscript by Feb 27 2020 11:59PM. Please email us (plosmedicine@plos.org) if you have any questions or concerns.

Please let me know if you have any questions. Otherwise, we look forward to receiving your revised manuscript in due course. 

Sincerely,

Richard Turner PhD, for Thomas McBride, PhD

rturner@plos.org

Please adapt the data statement to state "... because of constraints dictated by the study's ethics approval ..." or similar. 

Please adapt the title to include a study descriptor and accord with journal style. We suggest "Healthy behaviors at age 50 years and frailty at older ages in a 20-year follow-up of the Whitehall II cohort: a longitudinal study". 

Please add summary demographic information on participants to the abstract. 

In the abstract and elsewhere, please add p values alongside 95% CI where available. 

Please add a new final sentence to the "methods and findings" subsection of your abstract to summarize the study's main limitations. 

At line 45, please begin the sentence with "Our findings suggest that ..." or similar. 

After the abstract, we will need to ask you to add a new and accessible "author summary" section in non-identical prose. You may find it helpful to consult one or two recent research papers published in PLOS Medicine to get a sense of the preferred style. 

Early in the methods section of your main text, please state whether the study had a protocol or prespecified analysis plan and if so attach the document(s) as a supplementary file (referred to in the text). Please highlight analyses that were not prespecified. 

In the discussion of study limitations at line 281, please add some additional detail. For example, in noting the limitations of observational evidence, you could mention the possible existence of unmeasured confounders. 

Throughout the text, please style reference call-outs as follows: "... involved in frailty development [49,50]." (i.e., preceding punctuation). 

In the reference list, please ensure that journal names are abbreviated correctly and consistently (e.g., "Lancet" for references 2, 4 and 8). 

Please additional access details to reference 10. 

Please add a completed checklist for the most appropriate reporting guideline, which we suspect will be STROBE, as a supplementary document (referred to in the methods section). In the checklist, individual items should be referred to by section (e.g., "Methods") and paragraph number rather than by page or line numbers, as the latter generally change in the event of publication. 

Comments from the reviewers:

*** Reviewer #1: 

"Association between healthy behaviors at age 50 and frailty at older ages: 20-year follow-up of the Whitehall II cohort study" analyzes the well-known Whitehall II cohort of British civil servants, in particular as to how "healthy behaviors" at age 50 affect "frailty" approximately 20 years later (with a range of plus or minus five years from the starting age, and also with a range for the follow-up depending on which of the four later clinical evaluations (2002, 2007, 2012, 2015) was most appropriate. The general findings were unsurprising, in that exhibiting more examples of healthy behaviours at age 50 correlated with reduced frailty later on. The hazard ratio for each additional healthy behaviour was estimated to be 0.69, and for all vs. no healthy behaviours as 0.28.

The authors used an appropriate interval-censored illness-death model for the main statistical analysis. Furthermore, the expected additional demographic and socioeconomic covariates were considered. On the whole, the analysis and conclusions seem broadly valid. However, there remain a few points that might be considered.

On the variability of the cohort participants in terms of initial age and follow-up age, it might be helpful to include more detailed data, such as a table with rows as initial age and columns as years to follow-up, with each cell containing the number of participants with that initial age/years to follow-up combination. Further on the data, while the demographic and healthy behaviour characteristics are provided in Table 1, there is no such information for frailty characteristics. This could be added.

On "Moderate alcohol consumption" being defined as a healthy behavior (line 95), the referenced source [32] states that "no robust association with alcohol consumption was found". In any case, it is unclear why "None" is not also defined as healthy behavior. As such, more involved analysis might be performed on alcohol consumption (e.g. comparing "None" only, and "None+Moderate" as criteria)

A general comment on the categorizations used for frailty and healthy behavior measures: while various justifications are cited (e.g. most prominently Fried's frailty phenotype for frailty), the availability of raw values for at least some of these measures suggests that more nuanced analyses might be possible for those measures at least. For example, the "slow walking speed" measure is suddenly discontinuous at a height of 173/159cm for males and females respectively. It could be interesting to examine effects with greater precision (e.g. for "Moderate alcohol consumption", the range of 1 to 14 units is also rather large and could conceivably be split further). Of course, it remains up to the authors as to whether these might constitute reasonable secondary analyses.

*** Reviewer #2: 

This original article evaluates the association between health behaviors at age 50 in Whitehall II cohort study participants and the incidence of the frailty syndrome, using Fried's phenotype, at 4 follow-up visits, in 2002, 2007, 2012 and 2015. The findings show that each healthy behavior was independently associated with lower risk of incident frailty, and that the impact of aggregate health behaviors was cumulative in further lowering the risk for frailty. 

Questions to authors:

Methods: p. 4: Baseline recruitment occurred in 1985-88, when those recruited were 35-55; it would be useful to know the range of length of follow-up after age 50 for those who were included in this analysis. Mean follow up length is mentioned on p. 16 as being 20 years.

Results: Given that p. 9 indicates confounding by higher SES and fewer comorbidities among those with greater number of healthy behaviors at age 50, it would be useful to understand if those with lower socio economic background who also had positive health behaviors had comparable outcomes to those of higher socioeconomic background. Further, is it possible to offer an analysis of length of exposure to adverse behaviors prior to age 50, by confounders?

p. 11: could the comparable incidence of frailty for moderate and high alcohol drinkers be associated with loss to mortality among the high drinkers?

p. 11: associations attenuated after adjustment for socioeconomic factors, comorbidities. Is the association of occupational position with frailty attenuated by healthy behaviors, and if so which?

Discussion:

It would be useful to hear analysis of the finding on p. 11 that there was no evidence of interaction between sex and number of healthy behaviors in association with frailty.

 p. 18. Lines 309-312. The aggregate health outcomes affected by these health behaviors warrants further discussion. Since those with positive health behaviors had fewer comorbidities at age 50, please discuss the findings re: independence of association of health behaviors and frailty from presence of chronic diseases, comorbidity.

*** Reviewer #3: 

Thank you for the opportunity to review the manuscript by Gil-Salcedo who investigated the association of health behaviors with developing physical frailty in adults who were 50 years old at baseline. Midlife represents a critical time in which healthy lifestyle behaviors could play a major role in helping to prevent the health problems of old age (e.g., dementia PMID: 28735855) and understanding health behaviors may be pivotal in frailty prevention (PMID: 26805753). In this study, participants from the Whitehall II cohort study (n= 6357) had their smoking status, alcohol consumption, physical activity, and fruit/vegetable consumption measured at their first study visit (at age 50). Frailty was assessed at four time points, over a period of 20 years. All health behaviors were independently associated with developing frailty (445/6357 developed frailty) even when all health behaviors were included in their model which considered death as a competing risk. Non-smokers had the lowest risk of developing frailty, followed by moderate alcohol consumption, physical activity, and then fruit/vegetable consumption. A greater number of health behaviors resulted in a dose-associated lower risk of frailty. 

Major comments:

-Have the authors considered examining sedentary time in their models? Sedentary behaviors (low energy expenditure activities while seated/lying/reclining during awake hours) are associated with frailty (PMID: 30355522) and may provide insights into their data. It might be possible that even though someone is physically active, they could still report high levels of sitting time.

- A novelty of this study is the examination of the impact of multiple health behaviors on developing frailty. However, it is reasonable to assume that health behaviors changed over the follow up period in the present study, and such changes are likely to impact whether an individual develops frailty or not. Based on the author's description of the Whitehall II study, it is feasible to assess changes in health behaviors. It is strongly recommended that changes in health behaviors be assessed in relation to developing frailty, as these results would be quite novel.

-The authors adjust for several chronic conditions at age 50 in their statistical models, some of which are more severe than others. Can the authors determine which baseline conditions were more likely to result in developing frailty? 

Minor comments:

-It appears that disability was not considered in their models. How might they affect the findings here? Were participants free of disability at baseline?

-Can the authors please comment on the multiple frailty tools which exist? Although the frailty phenotype is among the most widely used in research, how might the findings differ if another frailty tool, say the frailty index based on the accumulation of deficits model, were used?

-It is recommended that the authors should comment on the implications of their findings relation to public health and in the clinical context.

***

[LINK]

---

## [Decision Letter · Decision Letter 1]

1 May 2020

Dear Dr. Sabia,

Thank you very much for re-submitting your manuscript "Healthy behaviors at age 50 years and frailty at older ages in a 20-year follow-up of the Whitehall II cohort: a longitudinal study" (PMEDICINE-D-19-04222R1) for review by PLOS Medicine.

I have discussed the paper with my colleagues and the academic editor and it was also seen again by two reviewers. I am pleased to say that provided the remaining editorial and production issues are dealt with we are planning to accept the paper for publication in the journal.

[LINK]

We look forward to receiving the revised manuscript by May 08 2020 11:59PM. 

Sincerely,

Thomas McBride, PhD

Senior Editor 

PLOS Medicine

plosmedicine.org

Requests from Editors:

1- Please consider addding "UK" to the title: "Healthy behaviors at age 50 years and frailty at older ages in a 20-year follow-up of the UK Whitehall II cohort: a longitudinal study"

2- In the "Study population" section of the Methods, please specify whether consent is written.

3- In the Discussion, please switch the order of the “Strengths and limitations” and “Comparison with previous studies” sections.

4- Regarding your response to reviewer 2, point 5, it may be relevant to mention this point and reference the Brunner study in the Discussion.

Comments from Reviewers:

Reviewer #1: We thank the authors for carefully considering our previous comments, and performing appropriate post-hoc analyses. The remaining minor suggestions pertain mostly to the supplementary material:

1. There appears to be an issue with the image for S3 Figure (green background, blurred words)

2. For Table 2/3 & S4 to S7 Tables, it might be helpful for the N frail/N total ratio to also be expressed as a percentage for easy comparison, if formatting permits.

Reviewer #3: The reviewers have done a great job addressing reviewer comments. I accept the article in its revised form.

[LINK]

---

## [Editor Report · Decision Letter 2]

11 Jun 2020

Dear Dr. Sabia, 

On behalf of my colleagues and the academic editor, Dr. Sanjay Basu, I am delighted to inform you that your manuscript entitled "Healthy behaviors at age 50 years and frailty at older ages in a 20-year follow-up of the UK Whitehall II cohort: a longitudinal study" (PMEDICINE-D-19-04222R2) has been accepted for publication in PLOS Medicine. 

PRODUCTION PROCESS

PRESS

PROFILE INFORMATION

Thank you again for submitting the manuscript to PLOS Medicine. We look forward to publishing it. 

Best wishes, 

Thomas McBride, PhD

Senior Editor 

PLOS Medicine

plosmedicine.org